# A Study on Miniaturized In-Situ Self-Calibrated Thermometers Based on Ga and Ga-Zn Fixed Points

**DOI:** 10.3390/s24175744

**Published:** 2024-09-04

**Authors:** Haiying Huang, Wenlu Cai, Yongjian Mao, Kun Wan, Yong Wen, Yuqiang Han, Qiang Zhang, Rong Zhang, Xing Zheng

**Affiliations:** Institute of Systems Engineering, China Academy of Engineering Physics, Mianyang 621999, China

**Keywords:** thermometer, in-situ calibration, self-calibration, Ga fixed point, Ga-Zn fixed point

## Abstract

In order to ensure the reliability and accuracy of long-term temperature measurement where the thermometers are discommodious or even impossible to access for conventional periodical calibration, a study on miniaturized in-situ self-calibrated (MISSC) thermometers based on Ga and Ga-Zn fixed points was conducted using temperature scale transfer technology. One MISSC thermometer consists of three parts: the first is the fixed-points hardware, including a container with two cells separately filled with Ga and Ga-Zn; the second is the temperature sensing hardware, made of a Type T thermocouple; the third is the mini-power heating hardware, made of a film resistance. The measurement and calibration (M&C) system comprises a temperature measurement and data processing subsystem and a mini-power heating control subsystem. Then, an in-situ self-calibration can be carried out by mini-power heating from a room temperature of about 20 °C, and then by comparison between the measured phase transition plateau results and the standard fixed-points, i.e., Ga fixed point (about 29.76 °C) and Ga-Zn fixed point (about 25.20 °C). A series of experiments were performed, and the results show that: (1) both the proposed hardware design and the self-calibration method are feasible, and (2) the Φ16 mm × 25 mm MISSC thermometer is found to be the most miniaturized one that can realize reliable self-calibration in this study.

## 1. Introduction

It is well known that temperature measurement is very widely used and that precision is increasingly urgent to improve in various fields such as scientific research and industrial production. In some vital measurements, thermometers are required to be periodically taken out and calibrated using a comparison method to guarantee reliability and accuracy. However, there are some specific situations where the thermometers are non-detachable and non-accessible during long-term temperature measurements, and thus discommodious or even impossible to be taken out for conventional periodical calibration. Therefore, in-situ calibration techniques are indispensable to be developed in those situations.

Up to now, the in-situ self-calibration technology has caused wide public concern and a number of studies have been performed. Rao et al. [1] developed a self-calibrated luminescent temperature sensor with relatively high absolute and relative sensitivity using a highly sensitive mixed lanthanide metal-organic framework. Wang et al. [2] introduced a dual-emitting Cu6-Cu2-Cu6 cluster as a self-calibrated and wide-range luminescent molecular thermometer. Takei et al. [3] explored a nanoparticle-based ratio metric and self-calibrated fluorescent thermometer for intracellular temperature measurement in real time. Chen et al. [4] designed a highly sensitive dual-phase nano-glass-ceramics self-calibrated optical thermometer via the usage of Ln^3+^ (Ln = Eu, Tb, Dy) luminescence as the reference signal and Cr^3+^ emission as the temperature signal. Ding et al. [5] described a new non-contact self-calibrated optical thermometer based on the Ce^3+^→Tb^3+^→Eu^3+^ energy transfer process, which provides a promising approach to designing a non-contact self-calibrated optical thermometer with high-temperature sensitivity and good signal discriminability. Wu et al. [6] synthesized the Pr^3+^/Tb^3+^ doped LuNbO_4_ phosphors and then developed a self-calibrated optical thermometer based on the intervalence charge transfer transitions. Chen et al. [7] synthesized the Bi^3+^, Eu^3+^co-doped SrLu_2_O_4_ phosphors and then designed a self-calibrated optical thermometer. Bao et al. [8] proposed a multimode fiber-optic surface plasmon resonance temperature sensor based on self-calibrated dual-channel sensing, and experimentally demonstrated the sensing performance of the structure. Wan et al. [9] presented a dye-loaded nonlinear metal-organic framework as a self-calibrated optical thermometer. Li et al. [10] explored a highly sensitive blue-LED-excitable self-calibrated luminescent thermometer based on Cr^3+^/Eu^3+^ Co-doped Al_2_W_3_O_12_. It can be seen that most of the mentioned studies [1,2,3,4,5,6,7,9,10] refer to optic, luminescent, or non-contact temperature sensing.

As for electric thermometers, such as thermocouples and platinum resistances, the in-situ self-calibration methods have not been well investigated. Much attention has been directed toward the calibration methods by using fixed points techniques in recent decades, especially since the International Temperature Scale of 1990(ITS-90) [11] was published. As the reference point of the international unit, the phase transition fixed point has excellent characteristics, such as good temperature reproducibility and low uncertainty, and was a significant technical way to solve the demand for high-precision temperature calibration. Burdakin et al. [12,13] discussed the melting/freezing curves for the single component Ga and bimetallic eutectic alloys Ga–In, Ga–Sn, Ga–Zn and Ga–Al in small-sized cells. The length of the cell is 75 mm, and the diameter is about 30 mm. Ivanova et al. [14,15] studied the fixed points based on Ga-In and Ga-Zn eutectic alloys. Marin et al. [16] studied the melting curves and temperature filed of a multiple fixed-point cell using thermal finite element simulations. Ongrai et al. [17,18] developed a multi-mini-eutectic cell with high-temperature Fe–C and Co–C fixed points for Type C thermocouple self-calibration. The cell contains two layers of different eutectic materials in the same crucible, one in each compartment, separated by a thin graphite disk. The total length of the cell was 18.5 mm, and the diameter was 10 mm. Kim et al. [19] explored a fixed-point cell that contains multiple metal elements (aluminum, silver and copper), and the results confirm the suitability of metal multicells for the calibration of secondary thermometers. Ragay-Enot et al. [20] designed and fabricated a mini multi-fixed-point cell (length 118 mm, diameter 33 mm) containing three materials (In-Zn, Sn and Pb) in a single crucible for the easy and economical fixed-point calibration of industrial platinum resistance thermometers for use in industrial temperature measurements. Hao et al. [21] studied the phase-change plateaus of the Ga and Ga-In alloy fixed points, in which the packaged Ga is 4.450 g. Sun et al. [22] performed an onsite calibration of a precision industrial platinum resistance thermometer of approximately 6 cm diameter and 8 cm length near room temperature based on a series of small-size eutectic points, including Ga-In (15.7 °C), Ga-Sn (20.5 °C), Ga-Zn (25.2 °C) and a Ga fixed point (29.7 °C), which was developed in a portable multi-point automatic realization apparatus. Elliott et al. [23] investigated the robustness of the self-validation method by constructing and testing two different designs of a self-validating device over 60 h to 90 h. Edler et al. [24] tested the self-validation concept for thermocouples to monitor their performance in the temperature range between 1000 °C and about 1800 °C using integrated miniature fixed-point units and demonstrated its suitability to provide long-term confidence in industrial high-temperature measurements within about (2–3) K. From the above-mentioned studies, it can be found that fixed-point cells have been explored and applied in calibrating thermometers, but the dimensions of the fixed-point cells are relatively large and they are difficult to use in self-calibration applications; moreover, few integrative in-situ self-calibrated thermometers were developed.

Aiming to develop a miniaturized in-situ self-calibrated (MISSC) thermometer using near room temperature, three prototype thermometers were designed based on Ga and Ga-Zn fixed points with different dimensions. Their properties were investigated and compared in order to select the most miniaturized dimension with excellent in-situ self-calibration performances.

## 2. Self-Calibration Method

The fixed-point materials Ga metal and Ga-Zn alloy are both in a solid state under room temperature conditions (about 20 °C). During the mini-power heating process, Ga-Zn alloy and Ga metal will undergo a solid–liquid phase transition, respectively. In the phase transition processes (solid–liquid mixed state), the phase transition temperature plateaus will be generated, and the temperature values corresponding to the plateaus are the melting points of the fixed-point materials. Theoretically, the Ga-Zn fixed point is 25.20 °C and the Ga fixed point is 29.76 °C, referring to ITS-90. By measuring the temperature of the whole process with a sensor (such as thermocouple or platinum resistance), the temperature sensor could be calibrated by using the phase transition temperature plateaus of the fixed points. The application flowchart of the MISSC thermometers is illustrated in Figure 1, from which it can be seen that an initial calibration process should be carried out for both fixed points and the temperature sensor before installation, and one or more in-situ periodical self-calibrations should be conducted after installation.

## 3. Hardware Fabrication & Integration

The related hardware consists of two parts: one is the miniaturized in-situ self-calibrated (MISSC) thermometers, and the other is the temperature measurement and calibration (M&C) system.

### 3.1. Fabrication of MISSC Thermometers

The MISSC thermometer is designed expertly and the typical schematic diagram is depicted in Figure 2a. A MISSC thermometer mainly includes three components:A temperature sensing component. The copper-constantan (type T) thermocouple manufactured by Chengdu Shuangtie Instrument Co., Ltd. (Chengdu, China) was chosen as the temperature measurement sensor. It also can be replaced by other sensors such as platinum resistance;A fixed point container. It is a cylindrical soaking block shell with two fixed point material cells, separately filled with Ga and Ga-Zn. The cylindrical soaking block shell was made of graphite to guarantee a uniform temperature field;A flexible heating resistance film. It was uniformly pasted and wrapped around the cylindrical soaking block shell and could be used to heat the fixed point container when self-calibration was carried out. For safety considerations, a fuse is adopted and connected between the film and the power wire. An asbestos heat insulation layer was wrapped on the outermost wall.

Three different dimensions of MISSC thermometers were fabricated and assembled, as shown in Figure 2b. The detailed information is listed in Table 1.

### 3.2. Integration of M&C System

For both experimental and practical applications, a system that combined the functions of both measurement and self-calibration, namely the M&C system, was established, as shown in Figure 3. The integrated M&C system comprises three functional parts:A heating control subsystem. It mainly includes a direct current (DC) power (DH1799M) made by Beijing Dahua Radio Instrument Co., Ltd. (Beijing, China) and a renewable fuse, which are used to control the heating power and ensure the temperature rising rate of the fixed-point materials.A temperature measurement subsystem. It is adopted to measure the temperature according to the temperature-sensing component in the MISSC thermometer. In detail, it consists of two parts: one is the temperature transmitter that can transmit the weak electromotive force signal generated by the thermocouple to a normalized voltage signal; the other is the data acquisition system whose hardware and software were installed in a personal computer (PC).A data processing subsystem. It is a self-developed software installed in the PC that is used to collect and process the temperature data obtained during mini-power heating for in-situ self-calibration and presenting a calibration result.

## 4. Experimental Results and Discussion

### 4.1. Initial Calibration

Referring to the application flowchart shown in Figure 1, the three MISSC thermometers listed in Table 1 were initially calibrated by comparison method with a high-precision type T thermocouple from a professional calibration laboratory, both for the fixed points and the temperature sensors. The constant current of 0.06 A was applied to both the No.1 and No.2 fixed points, while 0.08 A was chosen for the No.3 fixed point. The related parameters exploited for the heating control subsystem are presented in Table 2. The results of the three fixed points were presented in Figure 4, Figure 5 and Figure 6, respectively, which were measured by the pre-calibrated temperature testing system with the extended uncertainty of 0.05 °C (*k* = 2). From the figures, it can be apparently seen that the phase transition temperature plateaus lasted a quite long period, enough to conduct self-calibration, with the temperature changing on a small scale. What should be emphasized is that two-phase transition temperature plateaus for Ga-Zn and Ga fixed points should appear correspondingly. The measured average temperature values of 25.27 °C and 29.77 °C for No.1 fixed points and 25.21 °C and 29.81 °C for No.2 fixed points were determined by calculating the average temperature of the phase transition plateaus, which would be adopted to produce self-calibration for the temperature sensors. However, only one phase transition temperature plateau was presented by Ga solid to liquid transition for the No.3 fixed point, and the reason may be that the Ga-Zn alloy filled in the No.3 MISSC thermometer was in a liquid state at the room temperature of 20 °C due to the supercooling phenomenon. It is a unique phenomenon occurring for some liquid metals, especially for the Ga and Ga-based alloys, which mainly presents the solidification point as sensitive and changing due to several conditions, such as superheating or supercooling the fixed points, non-uniformity of the crystal structure, the effect of heat exchange conditions with the surroundings and so on [15,25,26,27]. A detailed discussion will be given in Section 4.3. Furthermore, the extended uncertainty of six thermocouples used for self-calibration in this paper was pre-calibrated and estimated at 0.2 °C (*k* = 2), where the uncertainty of the temperature measuring equipment, data recording device and the calibration standard source have been considered, which was carried out by the same professional calibration laboratory.

### 4.2. Self-Calibration

After the initial calibration, self-calibrations were carried out for No.1 and No.2 MISSC thermometers to verify the developed design and calibration method. The main parameters of the heating procedure were the same as those in the initial calibrations. What should be noticed was that the temperature was also measured and collected in the cooling period to ensure and validate that the Ga and Ga-Zn in the MISSC thermometers can solidify in the laboratory at room temperature. For the No.1 MISSC thermometer, the heating started from the temperature of 18.75 °C and lasted for 18,730 s when the temperature reached 33.66 °C and then naturally cooled to the laboratory temperature. During the whole process, three type T thermocouples that had been pre-calibrated, as stated in the initial calibration part, were located in three areas for temperature monitoring/measurement, i.e., the outer wall of the Ga fixed-point cavity, the outer wall of the gallium–zinc alloy fixed point cavity and the sensor mounting block, to illustrate the temperature uniformity of the whole fixed point container, which was demonstrated with the three partly zoomed-in measured temperature curves nearly coinciding, as shown in Figure 7. The average temperature values of melting phase transition plateaus for Ga-Zn and Ga materials in the heating procedure were calculated as 25.39 °C and 29.96 °C, respectively. Then, the self-calibration results could be achieved by making a difference in the average phase transition values from the self-calibration procedure and those from the initial calibration procedure, i.e., Ga-Zn (25.27 °C) and Ga (29.77 °C) solid–liquid phase transition. The errors of 0.12 °C and 0.08 °C were obtained as shown in Figure 8, which were in reasonable agreement with the extended uncertainty level of 0.2 °C of thermocouples. The self-calibration procedure described above was also conducted for the No.2 MISSC thermometer, and similar results were obtained, as shown in Figure 9. The average temperature values of the melting phase transition plateaus for Ga-Zn and Ga materials were 25.23 °C and 29.87 °C, respectively, from which the final self-calibration errors of 0.02 °C and 0.06 °C could be calculated by making comparisons with the initial calibration results of 25.21 °C and 29.81 °C, as shown in Figure 10. From the figures illustrated above, it can be concluded that it was feasible for the developed scheme and system to self-calibrate the temperature sensors with quite high precision.

### 4.3. Discussion

Aiming to discuss the influences of laboratory room temperature and heating power on the self-calibration results, the measured conditions were specially designed, and the experiments were accomplished. Firstly, the measurements were carried out in cases where different room temperatures were set as 18.77 °C and 20.25 °C, respectively. The results obtained in Figure 11 show that the solid–liquid phase transition temperature is almost same for both Ga-Zn and Ga fixed points, only with some difference in the duration of the phase transition plateaus. Secondly, various conditions of heating power were studied by changing the applied different current values. The constant currents of 0.08 A and 0.10 A were chosen to conduct the measurements. The tested temperature curve was plotted in Figure 12, where it can be seen that the phase transition duration decreased significantly, and the temperature values changed to a certain extent. Therefore, it suggests that the heating power should be assigned to the unique MISSC thermometer during the whole service period, i.e., both for the initial calibration and self-calibration. Finally, a specially designed experiment verified that the reason that there was no appearance of Ga-Zn phase transition plateaus during the heating procedure for No.3 MISSC thermometer was super cooling. The main procedure could be described as “5 °C cooling->room temperature->heating->room temperature cooling” and the temperature history curve was plotted as shown in Figure 13. It can be noted that the liquid–solid phase transition began at the container temperature of 10 °C, with the plateaus lasting about 800 s, then the MISSC thermometer cooled to 5 °C. After that, the MISSC thermometer was placed in the conventional laboratory environment (19.7 °C) for 720 s, and the heating current was applied to the resistance film to raise the temperature to 35.5 °C of the fixed point. As can be expected, the two stages of the solid–liquid phase transition plateaus were presented for Ga-Zn and Ga materials. During the room temperature cooling procedure, just the Ga liquid–solid phase transition was achieved. The results testified to the assumption of the supercooling phenomenon of the No.3 MISSC thermometer, thus it cannot be adopted to calibrate the temperature sensor at 25.20 °C with the Ga-Zn fixed point.

## 5. Conclusions

In this paper, the integrated temperature measurement and self-calibration scheme based on fixed points was put forward and three MISSC thermometers with various dimensions were particularly designed and fabricated. The experiments were accomplished to gain the temperature data, from which it was validated that the in-situ self-calibration in temperature values of 25.20 °C and 29.76 °C can be achieved for No.1 and No.2 MISSC thermometers, respectively. It confirmed the feasibility and applicability of the fabricated fixed points for the calibration of temperature sensors. The container of the fixed point could not be too miniaturized because the supercooling phenomenon of the Ga-Zn alloy and the smallest size can be regarded as Φ16 mm × 25 mm. Further detailed research regarding supercooling will be conducted in the future, mainly via numerical simulations and experimental verifications.

## Figures and Tables

**Figure 1 sensors-24-05744-f001:**
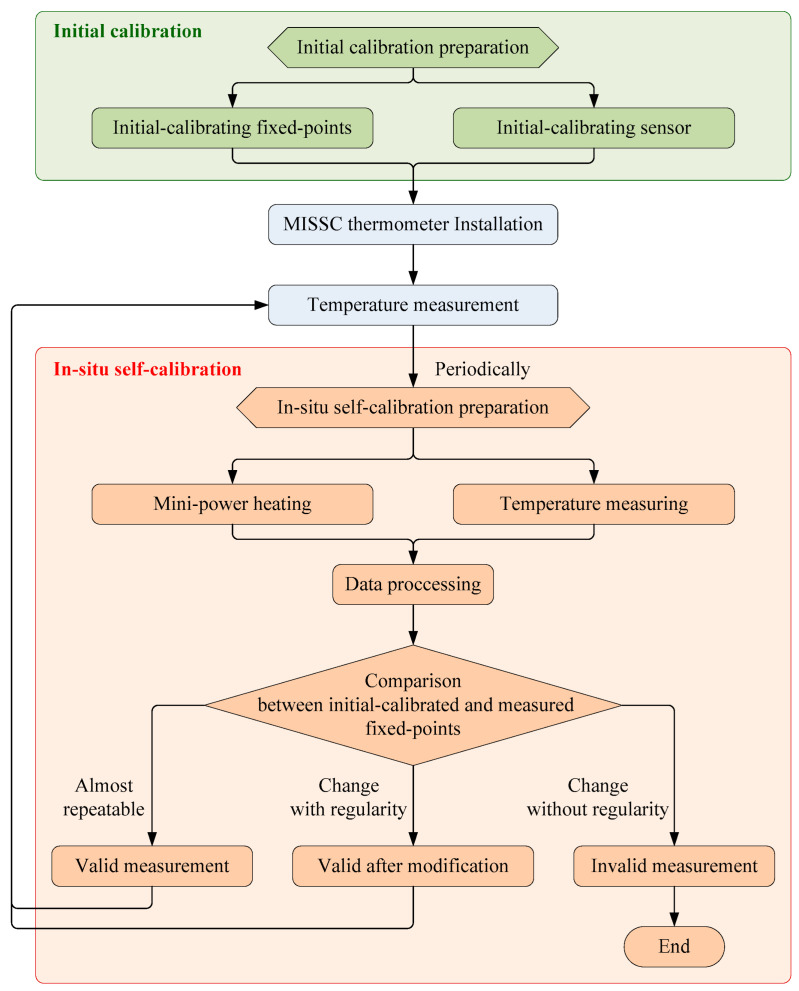
The application flowchart of MISSC thermometers.

**Figure 2 sensors-24-05744-f002:**
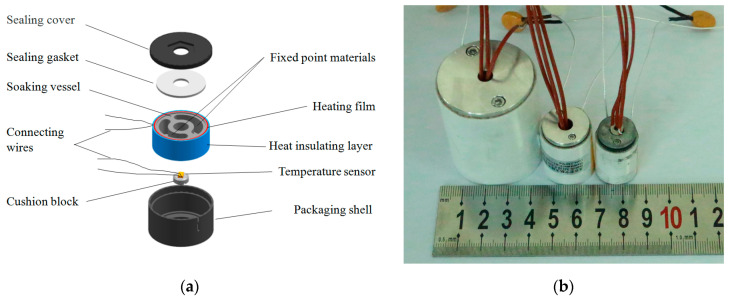
MISSC thermometer. (**a**) Schematic diagram; (**b**) Photo.

**Figure 3 sensors-24-05744-f003:**
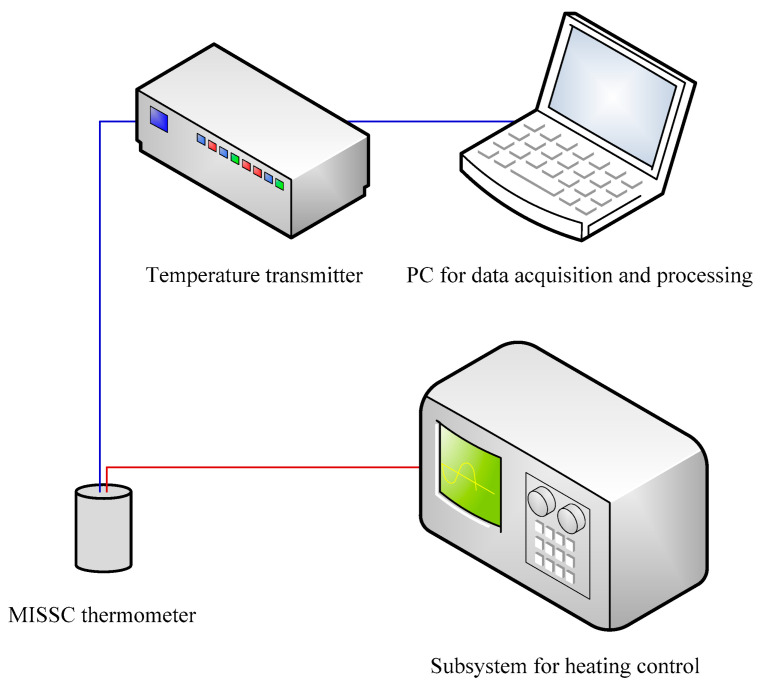
Integrated M&C system.

**Figure 4 sensors-24-05744-f004:**
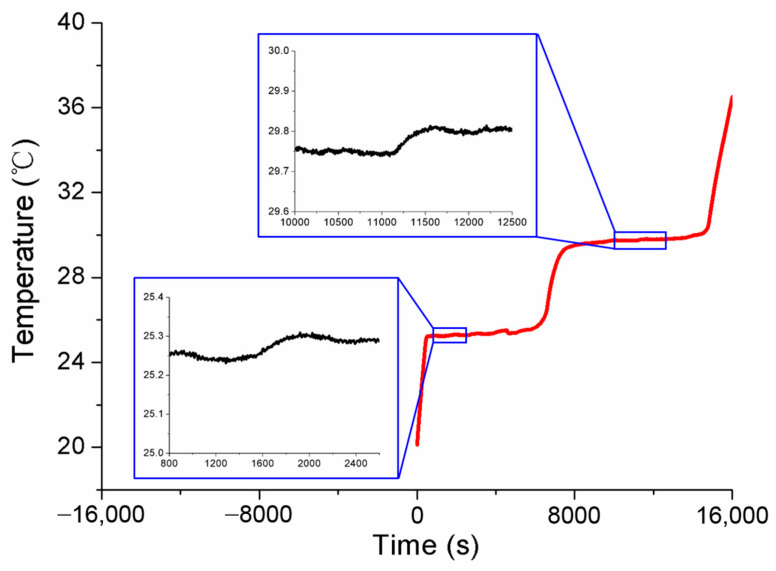
Temperature history of initial calibration for the fixed points of the No.1 MISSC thermometer.

**Figure 5 sensors-24-05744-f005:**
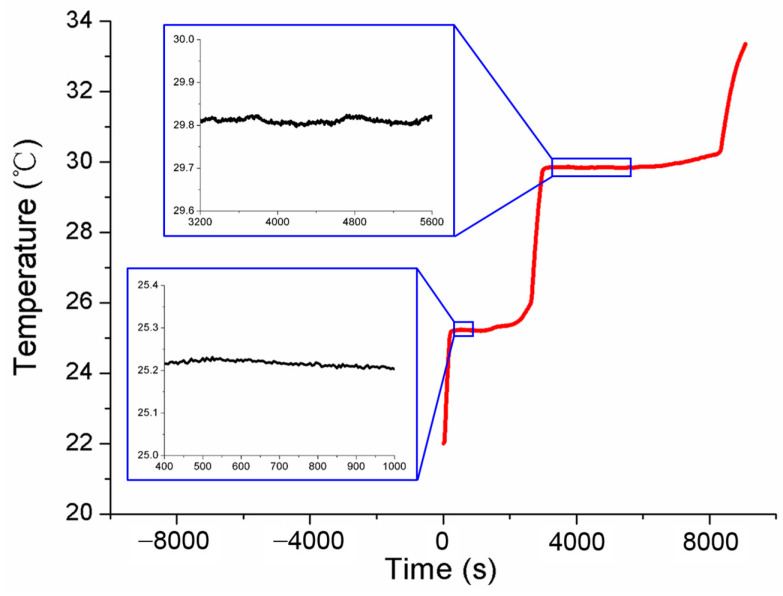
Temperature history of initial calibration for the fixed points of the No.2 MISSC thermometer.

**Figure 6 sensors-24-05744-f006:**
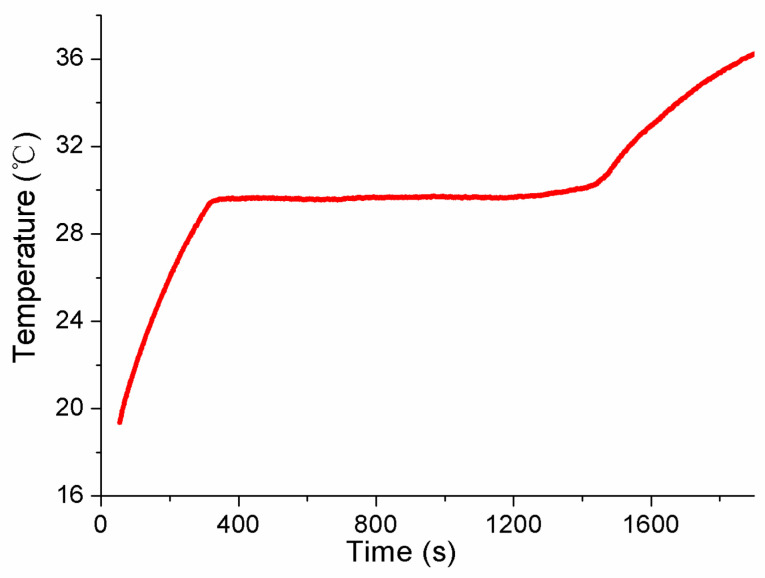
Temperature history of initial calibration for the fixed points of the No.3 MISSC thermometer.

**Figure 7 sensors-24-05744-f007:**
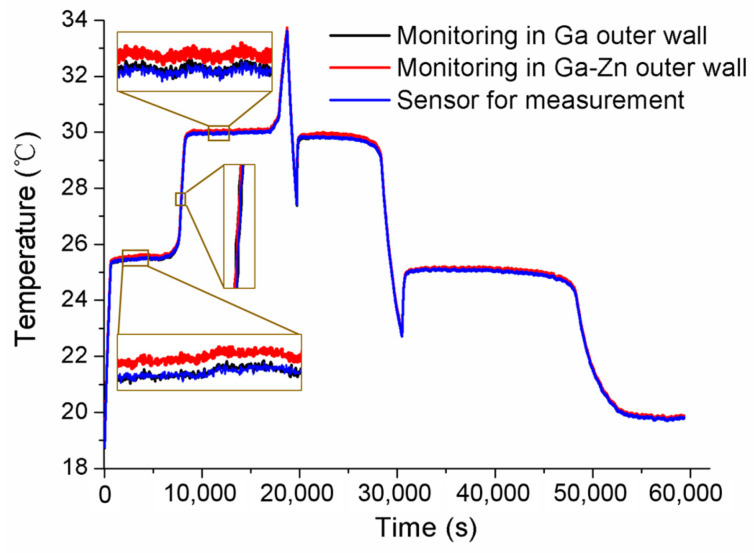
Temperature histories of self-calibration for the No.1 MISSC thermometer.

**Figure 8 sensors-24-05744-f008:**
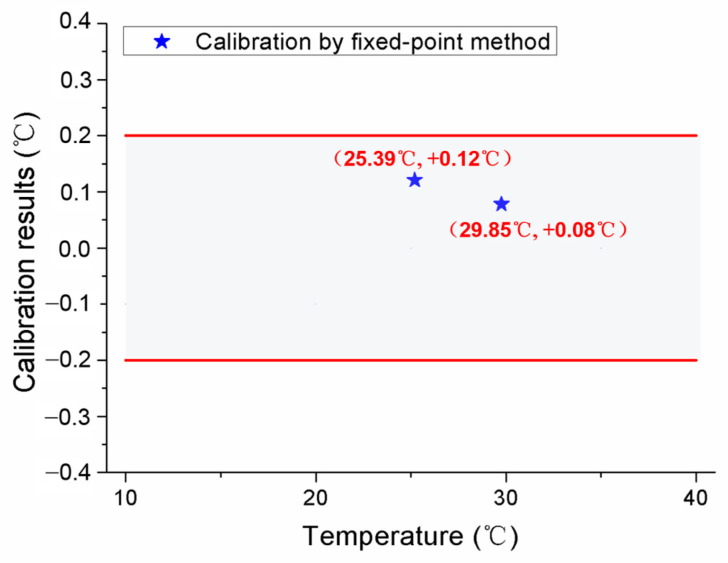
Self-calibration results for the No.1 MISSC thermometer.

**Figure 9 sensors-24-05744-f009:**
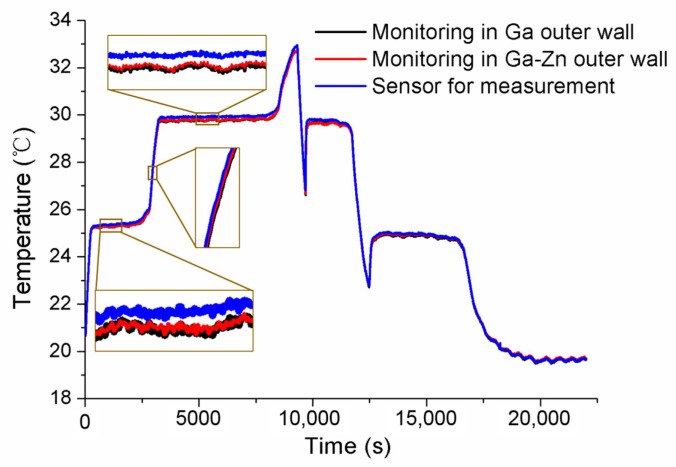
Temperature histories of self-calibration for the No.2 MISSC thermometer.

**Figure 10 sensors-24-05744-f010:**
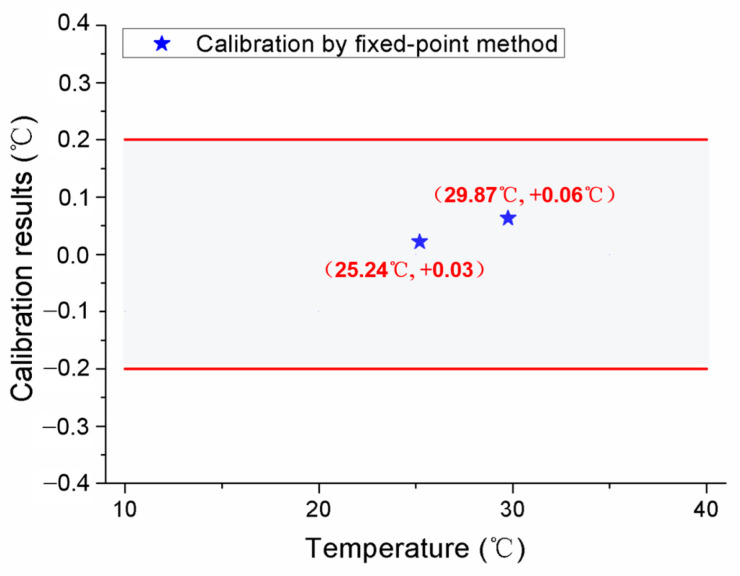
Self-calibration results for the No.2 MISSC thermometer.

**Figure 11 sensors-24-05744-f011:**
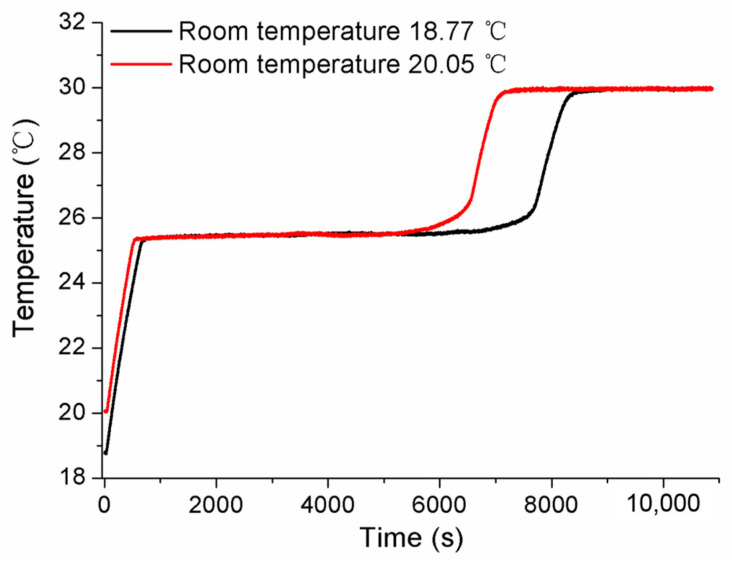
Influence of room temperature on measured fixed points for the No.1 MISSC thermometer.

**Figure 12 sensors-24-05744-f012:**
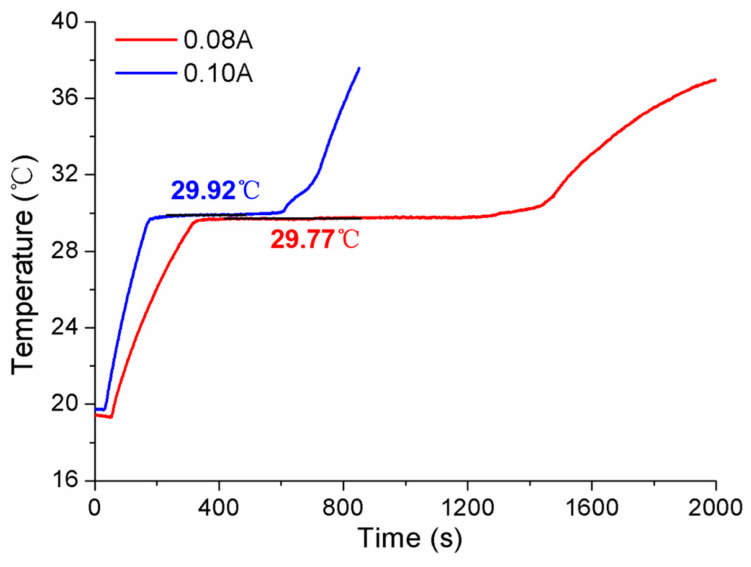
Comparison of results under various heating power for the No.3 MISSC thermometer.

**Figure 13 sensors-24-05744-f013:**
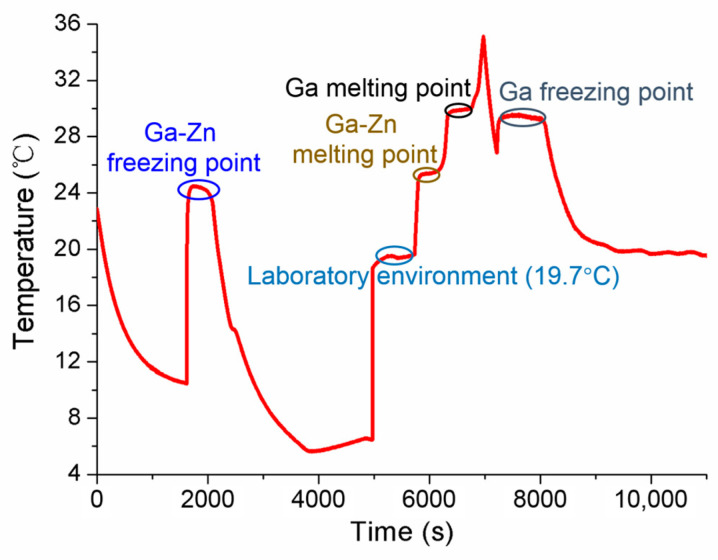
Verification of the super-cooling phenomenon for the No.3 MISSC thermometer.

**Table 1 sensors-24-05744-t001:** List of different dimensions of the three MISSC thermometers.

No.	Diameter/mm	Height/mm	Ga Mass/g	Ga-Zn Mass/g
1	40	50	86.44	87.21
2	20	30	8.87	8.92
3	16	25	4.04	4.18

**Table 2 sensors-24-05744-t002:** The resistance of heating film and heating parameters for the three MISSC thermometers.

No.	Resistance/Ω	Applied Current/A	Heating Power/W
1	455.9	0.06	1.64
2	128.1	0.06	0.46
3	83.6	0.08	0.54

## Data Availability

The data is unavailable due to privacy.

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
