# Peer review of "A Study on Miniaturized In-Situ Self-Calibrated Thermometers Based on Ga and Ga-Zn Fixed Points"

_sensors, 2024, doi:10.3390/s24175744_

Round 1
Reviewer 1 Report
Comments and Suggestions for Authors
The authors sholuld include additional references regarding miniature fixed points and self-validation.
There are no uncertainty discusssion, which is crucial for providing self-calibration.
Reviewer 2 Report
Comments and Suggestions for Authors
The manuscript compares 3 pairs of self-calibrated thermometers based on Ga and GaZn fixed points.
It presents a structure very appropriate to the work to be presented, describing well the thermometers and the measurement and calibration procedures. The results obtained are good and also include the study of the supercooling phenomenon.
The authors should correct and/or expand the following points to improve the quality of the manuscript.
1. The name figure 6 appears in two figures. Renumber in text and figure caption from that figure and the following ones.
2. Regarding the effect of super cooling, I believe it would be necessary to better explain the phenomenon to understand why it occurs in the smaller self-calibrated thermometer. Some references to the supercooling phenomenon could be included.
I recommend publishing the article with the exposed minor revisions.
Reviewer 3 Report
Comments and Suggestions for Authors
1. Figure 7 and 9 need to be better explained in the text. How were the errors calculated?
2. Figure 6 and 8: I suggest adding a zoomed-in inset to each figure to the plot so the black, red and blue lines are clearly seen
3.2Integration of M&C System
3. The type T thermocouple’s make (manufacturer) and model should be mentioned.
4. The the hearting control subsystem’s make (manufacturer) and model should be mentioned in the text.
5. What data processing subsystem software was used to process the temperature data?
6. Overall, the English language usage should be improved.
Comments on the Quality of English LanguagePlease seek help from a native English speaker to help with editing the manuscript.
